# Impact of Different Levels of Supervision on the Recovery of Severely Malnourished Children Treated by Community Health Workers in Mali

**DOI:** 10.3390/nu13020367

**Published:** 2021-01-26

**Authors:** Pilar Charle-Cuéllar, Noemí López-Ejeda, Mamadou Traore, Adama Balla Coulibaly, Aly Landouré, Fatou Diawara, Magloire Bunkembo, Antonio Vargas, Ruth Gil, André Briend

**Affiliations:** 1Action Against Hunger, C/Duque de Sevilla no. 3, 28002 Madrid, Spain; nlopez@accioncontraelhambre.org (N.L.-E.); avargas@accioncontraelhambre.org (A.V.); 2Doctorate Program in Epidemiology and Public Health, Rey Juan Carlos University, 28002 Madrid, Spain; 3EPINUT Research Group (ref. 920325), Complutense University of Madrid, 28040 Madrid, Spain; 4Nutrition Direction of the Ministry of Hygiene and Public Health, 1771 Bamako, Mali; douh35@yahoo.fr (M.T.); docteurada@gmail.com (A.B.C.); 5Institut National de Recherche en Santé Publique, 1771 Bamako, Mali; aland954@hotmail.fr (A.L.); diawarafatou@gmail.com (F.D.); 6Action against Hunger, 2562 Bamako, Mali; mbunkembo@ml.acfspain.org; 7Department of Public Health, Rey Juan Carlos University of Madrid, 28933 Madrid, Spain; ruth.gil@urjc.es; 8Department of Nutrition, Exercise and Sports, Faculty of Science, University of Copenhagen, 1870 Copenhagen, Denmark; andre.briend@gmail.com; 9Faculty of Medicine and Medical Technology, Center for Child Health Research, Tampere University, 33014 Tampere, Finland

**Keywords:** severe acute malnutrition (SAM), community health workers (CHW), integrated community case management (iCCM), supportive supervision, scaling-up interventions

## Abstract

(1) Background: The Ministry of Health in Mali included the treatment of severe acute malnutrition (SAM) into the package of activities of the integrated community case management (iCCM). This paper evaluates the most effective model of supervision for treating SAM using community health workers (CHWs). Methods (2): This study was a prospective non-randomized community intervention trial with two intervention groups and one control group with different levels of supervision. It was conducted in three districts in rural areas of the Kayes Region. In the high supervision group, CHWs received supportive supervision for the iCCM package and nutrition-specific supervision. In the light supervision group, CHWs received supportive supervision based on the iCCM package. The control group had no specific supervision. (3) Results: A total of 6112 children aged 6–59 months with SAM without medical complications were included in the study. The proportion of cured children was 81.4% in those treated by CHWs in the high supervision group, 86.2% in the light supervision group, and 66.9% in the control group. Children treated by the CHWs who received some supervision had better outcomes than those treated by unsupervised CHWs (*p* < 0.001). There was no difference between areas with light and high supervision, although those with high supervision performed better in most of the tasks analyzed. (4) Conclusions: Public policies in low-income countries should be adapted, and their model of supervision of CHWs for SAM treatment in the community should be evaluated.

## 1. Introduction

In Africa, 14 million children suffer from severe acute malnutrition (SAM), a form of malnutrition with an increased risk of death [1,2]. The use of appropriate prevention and management interventions for SAM could prevent 61% of SAM cases and almost 350,000 deaths per year globally [3]. To address these high rates of acute malnutrition, the community management of acute malnutrition (CMAM) approach was developed in the 2000s. This approach involves the timely detection of SAM in the community and outpatient treatment at health facilities (HFs) with ready-to-use therapeutic foods (RUTFs) for cases without medical complications [4]. In most low and middle-income countries, geographic and economic barriers to HF access are responsible for the low coverage of CMAM programs [5].

In Mali, the prevalence of global acute malnutrition, which is defined as a weight-for-height z-score (WHZ) of less than −2 or the presence of edema, is 10% (95% CI: 9.1–11.0), and the prevalence of SAM, which is defined as a WHZ of less than −3 or edema, is 2.0% (95% CI: 1.6–2.4), showing a serious situation. It is estimated that 34% of children’s deaths in Mali are related to malnutrition. The annual cost per child with malnutrition is estimated at 265.5 billion of CFA francs, corresponding to 4.06% of the gross domestic product [6].

The World Health Organization (WHO) supports universal health coverage, meaning that all individuals and communities should receive the health services they need without suffering financial hardship. The objective is to double health coverage by 2030 while ensuring the quality of services and cost-effectiveness of interventions [7]. To contribute to this target, WHO has made a strong recommendation to support the integration of community health workers (CHWs) into health systems as human resources that contribute to reducing infant mortality [8]. The results of a systematic review suggest that CHWs reduce inequity in health related to the place of residence, gender, education, and socio-economic status [9].

There is evidence regarding the effect that CHWs within the integrated community case management (iCCM) initiative have on the reduction in infant mortality related to malaria, diarrhea, and respiratory infections and on the treatment coverage of these diseases [10]. Evidence of how these CHWs can treat SAM and their effect on treatment coverage, effectiveness, and mortality is emerging. A review evaluating the effectiveness and coverage of SAM treatment with CHWs concluded that most of the interventions reached a cured ratio of approximately 90% [11]. These results have generally been achieved by small projects supported by non-governmental organizations (NGOs), and there is limited evidence at the regional or national level. Another systematic review of the use of middle-upper arm circumference (MUAC) by frontline workers to detect SAM cases concluded that the use of MUAC by CHWs increases the diagnosis and treatment coverage of SAM [12]. However, there is a need to identify factors that influence the performance of CHWs when scaling up this intervention [13].

In 2014–2016, a pilot study in Mali developed by the Direction de Nutrition of the Ministry of Health (MoH), the “Institut National de Recherche en Santé Publique” of Bamako, Action against Hunger (AAH), and The Innocent Foundation concluded that allowing CHWs to treat SAM reduces the defaulter ratio without compromising treatment outcomes and can lead to improved access to treatment [14]. This study showed that well trained and supervised CHWs are capable of managing cases of uncomplicated SAM and suggested that such a strategy could increase access to the quality of treatment in the country [15]. The intervention is cost-effective, and households receiving CHW-delivered care spent nearly half the amount of time and three times less money than those with SAM treatment received at an HF [16]. With this evidence, in 2015, the MoH of Mali included the treatment of SAM as part of the package of activities to be implemented by CHWs in his policy “Soins Essentiels dans la Communauté, Guide National de la Mise en Oeuvre” [17]. 

There is little evidence on the best approach to train and supervise CHWs in large-scale treatment programs integrated with other health activities. The objective of this research was to define the most effective level of supervision when scaling up SAM treatment using CHWs in Mali.

## 2. Materials and Methods

This was a prospective non-randomized community intervention trial with two intervention groups and one control group that compared the outcomes obtained with community treatment of SAM children under different levels of supervision.

### 2.1. Settings

This study was conducted from October 2017 to October 2018 in Mali in 3 different districts of the Kayes Region, including all the HFs and CHWs placed in those areas: Kita (50 HFs and 90 CHWs), Kayes (49 HFs and 45 CHWs), and Bafoulabé (21 HFs and 34 CHWs). Figure 1 summarizes all the study stages (A to I), highlighting activities that have been carried out differently in each of the three study groups. These activities are described in more detail below, indicating each case, which stage of Figure 1 corresponds to each activity.

A socio-economic cross-sectional survey was implemented before the intervention in September–October 2017 in all 3 areas to check their comparability (Figure 1A). The survey was administered to 1350 randomly selected households with a two-stage cluster sampling design: In the first stage, 30 villages were randomly selected in each of the 3 intervention groups, and in the second stage, 15 households were randomly selected within each village. Empty households, households without children, or households with children absent were not replaced. The head of the household or mother of the children was interviewed after their informed consent was obtained.

The baseline survey had different parts: (1) general information, such as date, name of the village, name of the health area, and household characteristics (e.g., family size, members by sex and age, and number of children under five); (2) living conditions, sanitation and hygiene infrastructure, and construction material of the house, socio-economic status, and the sources of income; (3) complementary feeding practices and calculation of the food consumption score (FCS) using the frequency of consumption of different food groups consumed by a household during the 7 days before the survey according to the World Food Programme guidelines [18]. This index considers the frequency of consumption of 9 food groups, each with a specific associated weight according to its nutritional importance. Poor diet diversification is defined by a score of 21 or less; (4) type of healthcare usually provided for children and the behavior of mothers/caregivers when the children are sick (presentation to a HF, traditional medicine, or self-medication); and (5) prevalence of acute malnutrition in children under the age of five diagnosed with a low MUAC.

### 2.2. Intervention

At the beginning of the study, all CHWs in each of the 3 groups received 21 days of training based on the package of iCCM using the training module of the MoH. The trainers were the health district management team together with AAH staff. This training included health promotion, infant and young child feeding (IYCF) practices, hygiene practices, family planning, neonatal care and management of diarrhea, malaria, pneumonia, and acute malnutrition. A pre and post-test was administered to all participants to ensure that the knowledge had been acquired correctly. As the time allocated to SAM, treatment in the training module seemed insufficient to ensure the quality of care, the study included a post-training internship for the CHWs at the HF level in the 3 study groups once a week for 6 consecutive weeks on the day of management of acute malnutrition at the HF. The objective of this internship was to reinforce the implementation of the appetite test, as-sessment of admissions and discharge criteria, and referral of children with complica-tions to inpatient care. Information related to socio-demographic characteristics was col-lected for all the CHWs during the training (Figure 1B).

### 2.3. SAM Management

All children who were 6–59 months old, presented to a HF or to a CHW’s site, and were diagnosed as having uncomplicated SAM were eligible for the study. Children identified at the community level as having SAM received treatment directly from the CHWs without being referred to a HF. The treatment of children with SAM by CHWs was progressively introduced during the first 3 months of the study, as CHWs were participating in the internship training at the HF level at the beginning of the study in November and December 2017.

Admission criteria to the program followed Mali’s National CMAM protocol: MUAC < 115 mm and/or WHZ < −3 based on the WHO growth standard and/or bilateral edema were used at HFs, but only MUAC and bilateral oedema were used at the CHW level [19]. In all 3 groups, clinical outcomes were evaluated. Curation was defined as a child with a WHZ ≥ −1.5 or MUAC ≥ 125 mm and absence of nutritional oedema during two consecutives visits. A defaulter case was defined as a child who missed two consecutive follow-up visits (14 days); a referred case was a case transferred to an inpatient care facility for treatment due to complications; and death referred to children who died during treatment.

Children with SAM received a weekly ration of RUTF of 170 kcal/kg/day until recovery. They also received systemic treatment with amoxicillin (50–100 mg/kg/day twice a day for five days) and one single dose of 500 mg of mebendazole at the first visit for deworming (Figure 1C).

### 2.4. Supervision of the Management of SAM in Different Districts

In the 3 study groups, the country’s CMAM and iCCM protocol was applied with different levels of supportive supervision. The control group was the district of Bafoulabé, where SAM treatment was delivered at the HF and CHW levels with the expected supervision recommended by the MoH but without any support from the AAH. In the Kayes district, light supervision was applied for which the AAH supervised the iCCM component of the program more closely. In the Kita district, high supervision was applied for which in addition to the close supervision of the iCCM activities, the CHWs also received monthly nutrition-specific supervision, and both were supported by the AAH.

The supervision period was from February 2017 to October 2018. Supportive supervision was planned to occur on a monthly basis and focused on identifying and solving problems and strengthening the health system from the community and included all the packages of the iCCM activities. As part of the supervision, the center’s technical director (DTC) had to complete a checklist and a booklet to assess the implementation of the recommendations made at each visit [17]. The checklist used in the iCCM activity supervision applied to the Kita and Kayes districts had 5 sets of questions: (1) clinical examination of the sick child with 13 different items collected; (2) newborn monitoring with 12 items; (3) family planning with 4 items; (4) IYCF with 3 items; and (5) hygiene and sanitation with 7 items (Figure 1D). The complete checklist is shown in the Appendix A.

In the Kita district, extra nutrition-specific supervision was implemented to assess the quality of care and performance provided by the CHWs, to identify training needs, to check the maintenance of equipment, to assess input and storage management, to check data collection and the quality of statistics, to assess the care of beneficiaries, and to identify implementation problems of SAM management. The checklist used during the nutrition-specific supervision applied in the Kita district was the same as the list included in the national CMAM protocol normally applied at HFs. This checklist has 8 set of questions: (1) Anthropometric and medical equipment; (2) Identification of danger signs; (3) Systematic screening; (4) Admission and discharge criteria application; (5) Appetite test performance; (6) Nutritional treatment; (7) Systematic medical treatment; and (8) IYCF promotion (Figure 1E). The complete checklist is shown in the Appendix A.

For each question, the performance of the CHW was classified as “passed” (1 point) or “failed” (0 points). The total score obtained from the sum of all the questions included in each set was recalculated over a maximum score of 10 points so that each set of questions could be scored between 0 and 10. A high quality of performance was considered when the total score was equal to or greater than 8 points.

### 2.5. Follow-Up and Monitoring Intervention Framework

Monthly meetings were held at the health area and district level with staff from the MoH (DTC of HF, CHWs, and nutrition focal point) and Action against Hunger’s staff (supervisors and physicians). The objective of these meetings were data monitoring and validation (Figure 1F,G). Two committees were set up at the beginning of the study. A technical committee responsible for providing expertise and advice on technical issues, tools, and materials and steering committee responsible of validation of the research protocol, approve the achievements and ensure compliance with the project’s strategic focus. Both committees were under the leadership of the MoH. They had a meeting every three months (Figure 1H,I).

### 2.6. Data Collection and Analyses

All the information was collected by AAH supervisors using the Open Data Kit or Microsoft Excel spreadsheets^®^ [20]. Disaggregated data from the admission and discharge of each child were collected from the registers of each HF or CHW’s site. In the two intervention groups of Kita and Kayes, the five sets of variables related to iCCM activities described above were collected during supervision on a monthly basis from direct observation of CHW performance using the iCCM checklist of the MoH. In the high supervision group of Kita, the 8 sets of variables related to nutrition activities described above were collected during the supervision every month using the checklist of the MoH for CMAM programs. For data analyses, the Kolmogorov–Smirnov test with Lilliefors correction was used to assess the normal distribution of continuous variables. A t-test or Mann–Whitney test was used to compare means or medians when appropriate. To compare discharge outcomes, chi-square tests were applied considering Yates’s correction when the minimum expected count for a category was five cases or fewer. All statistical analyses were performed with IBM® SPSS v.25 software.

### 2.7. Outcomes

The primary outcome of the study was the proportion of cured children in each district among those admitted with SAM at the HF or CHW level. Secondary outcomes were the proportion of defaulters, deaths, and references with complications and the quality of care provided by the CHWs during the iCCM supportive supervision and nutrition-specific supervision.

### 2.8. Ethical Approval

The study was approved by the Ethical Committee in Bamako (decision Nº 13/2017CE-INRSP). All the participants (mothers and children caregivers) were asked to sign a formal consent form before starting participation in the project. The study was registered in ISRCTN registry https://doi.org/10.1186/ISRCTN14990746.

## 3. Results

### 3.1. Baseline Characteristics of the 3 Study Areas

Out of the 1350 selected households, 130 were not included in the survey due to the absence of children in the household or an empty household. Information was collected from a total of 1220 houses: 412 in Kita, 407 in Kayes, and 401 in Bafoulabé. The main results of the survey are shown in Table 1. The areas were similar in terms of the number of children under the age of five in the families, children with acute malnutrition, and children with access to healthcare services. Some significant differences were found in living conditions, with a higher number of people having access to clean water, sand floor houses, and thatched-roof houses in Bafoulabé. By the other hand, this group is the one with poor dietary diversity. Regarding health care provision for sick children, no significant difference was found between the three groups.

The socio-demographic profile of the CHWs involved in the study is shown in Table 2. The level of school attendance was lower in Bafoulabé’s CHWs than in those from Kita and Kayes. More CHWs had more than 2 years of experience working as CHWs in Bafoulabé than in Kita and Kayes. Regarding the population covered in the catchment area, 13.3% of CHWs in the Kita group covered a population under 700 habitats, compared with Bafoulabé, where almost half of the CHWs, 41.1%, cover a population of 700 people. On the other hand, Kita’s group has a higher number of CHWs, 45%, who covered over 1500 inhabitants compared with the other two groups. CHWs in Bafoulabé and Kita were located further away from the HFs.

### 3.2. Outcome of Treatment in the Study Areas

The results on the effectiveness of SAM treatment are shown in Table 3. A total of 3320 children with uncomplicated SAM received outpatient treatment at the health structures in the Kita district (527 treated by CHWs and 2793 at HFs); 1930 children, at the health structures of Kayes (350 CHWs and 1580 HFs); and 862, in Bafoulabé (313 CHWs and 549 HFs).

The proportion of cured children was 81.4% in Kita, 86.2% Kayes, and 66.9% in Bafoulabé. The group without supported supervision, Bafoulabé, had a significantly lower proportion of cured children than the other groups. The Kayes district, the group that did not receive an extra nutrition-specific supervision, had the highest proportion of cured children. Bafoulabé is the only district that has not reached the Sphere Standard for humanitarian interventions set in 75% of children cured. The cure rates obtained by the CHWs in Kita district are significantly lower than those of the HFs despite having received nutrition-specific supervision, while no differences are found in Kayes’ district. In the case of Bafoulabé, the CHWs obtain even higher cure rates than the health personnel in formal health structures.

The proportion of defaulters was 6.4% in Kita, 1.8% in Kayes, and 15.3% in Bafoulabé with a non-significant difference between care providers in Kita but with better results with the CHWs in Kayes. In Bafoulabé, when we analyze the difference between care providers, CHWs had a defaulter ratio of 6.4% compared to 20.4% in the HF. When the three groups were pooled together, the defaulter ratio was 6.1%, with no significant difference between care providers.

The proportion of children transferred was 11.7%, 11.9%, and 16.9% in Kita, Kayes, and Bafoulabé, respectively. These figures include both those who developed a medical complication and those who did not properly respond to nutritional treatment, showing a reduction or stagnation in weight or the MUAC gain. The proportion of transferred children was significantly higher in Bafoulabé than in the other two districts. In Kayes, the proportion of transferred children was significantly lower with the CHWs, but the opposite was found for Kita and Bafoulabé, where it was significantly higher with the CHWs than with the HFs. The proportion of deaths was 0.4%, 0.1%, and 0.8% in Kita, Kayes, and Bafoulabé, respectively. Non-significant differences between care providers were found.

### 3.3. Supervision and Performance Scores of the CHWs

#### 3.3.1. iCCM Activities

There were no differences between the average number of supervisions for iCCM activities received by the CHWs of the Kita and Kayes districts (5.2 ± 1.8 vs. 4.6 ± 2.3, respectively, *p* = 0.105). However, in the high supervision area of Kita, there was a higher proportion of CHWs who received more than 5 supervisions during the nine months of the study from February to October 2018 (68.6% vs. 44.4% in Kita; *p* = 0.007). In contrast, in the district of Bafoulabé, which was not supported by the AAH, only 1 out of 35 CHWs was supervised during that period.

The results on the quality assessment of the CHWs’ performance on the iCCM activities in Kita and Kayes during the supervision are shown in Table 4. The CHWs from Kita had a significantly higher score for most of the tasks assessed for clinical examination of the sick child. Regarding nutrition items is remarkable: danger signs assessment CHWs in Kita have achieved a score of 9.85, evaluation of weight 9.57, temperature 9.87, measurement of the MUAC 9.59, breath movement assessment 9.64, use of the rapid diagnostic test for malaria 9.56, and correct triage they achieved a score of 9.58. However, most scores achieved by the CHWs of both groups were above 8 points for a maximum of 10.

In contrast, in the set of items related to IYCF, which was part of the iCCM package, the mean score was 6.20 in Kita and 7.18 in Kayes, and only 44.7% and 55.1% of CHWs in Kita and Kayes, respectively, obtaining a score of 8 or above. Detailed results by item can be found in Appendix A.

#### 3.3.2. Nutrition Activities

As mentioned previously, nutrition supervision was only implemented for the CHWs from the Kita district under high supervision. On average, each CHW was supervised 5.4 ± 1.9 times for the nutrition-specific activities during the nine-month supervision period. The average score obtained was over 8 points in all the sets of questions analyzed, except for the IYCF activities, where the obtained average score was 4.9 for a maximum of 10. The detailed results by item can be found in the Appendix A.

At the end of the study in October 2018, 100% of CHWs systematically screened all children for concomitant diseases, properly performed the appetite test, and correctly identified the danger signs. Oedema was correctly assessed by 98.9% of CHWs; 98.8% made a good identification of SAM children with colored MUAC tape; 98.4% correctly applied the admission and discharge criteria; 87.3% gave the needed systematic treatment; and 96.8% gave RUTF according to the national protocols. In contrast, the lowest score was achieved for IYCF promotion, and only 56.7% of CHWs provided nutritional counseling to pregnant women.

#### 3.3.3. Relationship between Supervision Activities and Treatment Outcomes

The association between the number of supervisions received by the CHWs and their quality of performance with the proportion of children cured after being treated for uncomplicated SAM is shown in Table 5. No significant difference was found between CHWs who received supervision above or below the median of 5 supervisions for the iCCM or nutrition-specific supervisions. The mean quality of performance score was obtained for all the CHWs, and the proportion of children cured in those reaching a score under and above the median was compared. Likewise, no significant difference was found for the iCCM or nutrition-specific supervisions. However, the median score was very high in both cases, with values above 9 over a maximum of 10.

The detailed results of the univariate linear regression analysis are shown in the Appendix A. None of the variables on the number of supervisions received or the quality of performance achieved in either the iCCM or nutrition-specific supervisions showed a statistically significant association with the percentage of children cured after treatment.

## 4. Discussion

This study examined the effects of different levels of supervision when decentralized treatment of SAM was scaled up at a regional or national level. It showed that supportive supervision resulted in improved outcomes compared with a non-supervised model, but there was no difference between areas with light and high supervision. The two groups with supervision met the Sphere standards. They obtained a cured ratio over 75% and a defaulter ratio less than 15%; in contrast to CHWs in the control group, the three groups obtained a mortality ratio less than 10% [21]. The performance of CHWs was higher in the high supervision district, but there was no relationship between the scores obtained from the tests and treatment outcomes.

To the best of our knowledge, this is the first study examining the effect of the level of supervision of CHWs delivering SAM treatment in a project being scaled up. Previous pilot studies have evaluated the effects of supportive supervision on clinical outcomes in reduced groups of CHWs but not in large scale projects. In general, these pilot studies suggested a positive impact of supervision on the performance of CHWs. Lazzerini et al. found that supportive supervision of CHWs significantly increased the number of children enrolled in the nutritional programs in Uganda [22]. In Ethiopia, the score obtained for service delivered by CHWs who received monthly supervision was higher than in the control group [23]. A study conducted in South Africa concluded that CHWs neglected the treatment of SAM without complications due to the lack of supportive supervision [24]. The synthesis of qualitative and quantitative assessment at the community level suggested that the lack of supervision was a challenge in the performance of CHWs in Madagascar [25]. In a systematic review, Ballard et al. identified supervision as a key intervention that is likely to improve the performance of CHWs in different health interventions not including the management of malnutrition [26].

Our results show a lack of impact of the level of supervision on treatment outcomes, which is consistent with the results of two reviews of non-nutritional programs, suggesting that the intensity of supervision had no major impact on the performance of CHWs. In a review of the impact of supervising CHWs in low-income countries, Zill et al. concluded that increasing the frequency of supervision did not necessarily lead to increased effectiveness, and the quality of supervision could play a more important role in the impact of this type of intervention [27]. In a systematic review about primary healthcare supervision, Bosch-Capablanch et al. also suggested that reducing the frequency of supervision could save costs, reducing supervisor salaries and travel cost, without a detrimental effect on performance [28]. However, another systematic review of the factors that affect the performance of CHWs showed that supervision increases the credibility and trust of families [29].

The lack of impact of extra monthly nutrition-specific supervision on the recovery of malnourished children could be explained by different factors. We suspect that many of the questions included in the evaluation grids were not adapted to the specific problems faced by CHWs, which are different from those faced by the health staff at HF-like structures, the organization of the space, reception and flow of patients, and staff responsibilities. Some items, such as the identification of danger signs during clinical examination, stock, and monitoring tools were already included in the supervision of the iCCM activities and may have had no additional impact on the outcomes as initially expected.

Our study did not analyze other actions performed during supportive supervision that could have had a positive effect on the performance of CHWs, such as the resolution of problems between the supervisor and the CHWs or the avoidance of problems with the supply of RUTF and systematic treatment. Furthermore, our study did not examine the effects of the salaries of CHWs. A systematic review by Rowe et al. suggests that economic incentives or integration within the formal health system may have an effect when several of these interventions are performed simultaneously.

To our knowledge, Mali is the first country in West Africa to make progress in scaling up SAM treatment with CHWs at the national level. Our results highlight the need to evaluate each component of a health and nutrition intervention, particularly, the supervision level. The technical committee of the project lead by the Directorate of Nutrition of the MoH and formed by representatives of the Regional and District Directorate of the MoH, the Regional Federation of Community Associations (FERASCOM), the World Food Programme, UNICEF, and other NGOs in addition to the AAH has made some recommendations based on the results of this study: (i) adaptation of the iCCM training module, including 2 days for acute malnutrition and the addition of a post-training internship at the health facility level; (ii) integration of nutrition supervision within the iCCM supervision activities and adaptation of the CMAM protocol supervision checklist at the CHW level; (iii) adaptation of the DTC supervision frequency of the CHWs so that the first 3 months when SAM treatment is integrated into the package of iCCM, CHWs will be supervised on a monthly basis and bimonthly supervision thereafter according to the performance and needs of the CHWs; and (iv) increase in the energy of coordination meetings at all levels of implementation (regional, district, health area, and village) with community leaders and community groups.

The work presented here has some limitations. This study was not a randomized control trial, and thus we could not rule out that our results were due to some extent to differences among districts independent of our intervention, as suggested by the survey of background socio-economic factors. The CHWs in the 3 districts started progressively treating children with SAM, depending on the availability of equipment. Therefore, the analysis we presented does not reflect the work of the 169 CHWs during the entire 10 months of the study, and some supervision checklists had missing data, which may limit our conclusions. The RUTF supply was ensured by the AAH, and no stock-out was registered, which means that these results may not be applicable in settings with RUTF stock-outs.

## 5. Conclusions

This study is the first to demonstrate in West Africa that supportive supervision of CHWs improves acute malnutrition treatment outcomes. Our results suggest that the decentralized service delivery model with CHWs may be one of the key solutions for policy makers to tackle malnutrition in Mali. We recommend following a supportive supervision model, including items related to the SAM management within the country’s existing CHW monitoring sheet, to carry out single joint supervision of all their activities. Likewise, it is advisable to follow up more closely on the service’s implementation set-up, but it could be spaced out to a bi-monthly follow-up after confirming that the treatment is being given with quality. Along with the supervision system, policymakers should ensure the supply chain of RUTF, the salary or motivation of the CHWs, and the health information system at the community level to make the decentralized SAM treatment a sustainable intervention. These policy makers, following the recommendations drawn from the study, can scale up treatment with CHWs in other regions of the country. Along with this intervention, the country should continue the analyses regarding how to ensure the chain supply chain and the health information system at the community level; and how these CHWs need to have their salary and motivation to make their work a sustainable intervention.

## Figures and Tables

**Figure 1 nutrients-13-00367-f001:**
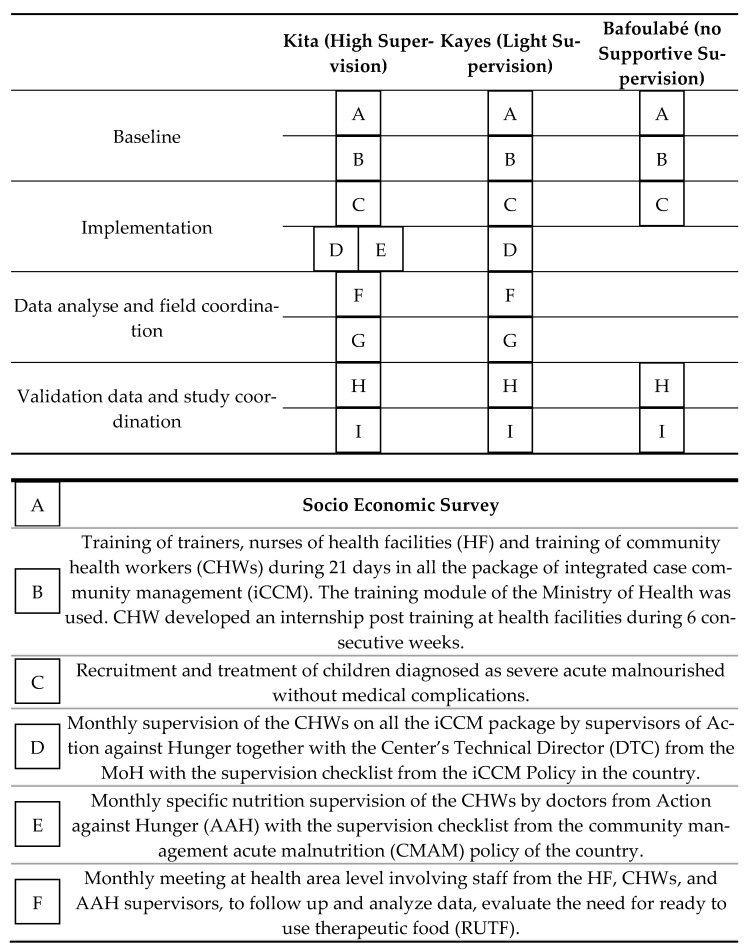
Intervention, monitoring, and supervision activities in the three groups of the study.

**Table 1 nutrients-13-00367-t001:** Baseline socio-economic characteristics in the three study groups.

Socio-Economic Item	Kita (High Supervision)	Kayes (Light SuperVision)	Bafoulabé (No SupPorted Supervision)	*p* Value
**Number of Surveyed Households**	412	407	401	
**Demographics**				
Sex of survey respondents (M/F)	1.12	1.06	1.14	
Age of survey respondents (Mean ± SD)	33.58 ± 12.56	34.24 ± 12.41	33.15 ± 15.91	0.046
Number of children under 5 years of age per household (Mean ± SD)	0.97 ± 0.86	1.00 ± 0.96	0.89 ± 0.89	0.087 ^NS^
Proportion of children aged 6–59 months with a MUAC ≤ 125 mm	M: 3 (1.4%)	M: 4 (1.9%)	M: 3 (1.5%)	0.905 ^NS^
	F: 6 (3.7%)	F: 4 (2.1%)	F: 5 (3.1%)	
**Living Conditions**				
Latrine available	406 (98.3%)	387 (95.1%)	395 (99%)	0.031
Clean water accessible	219 (52.9%)	169 (41.5%)	399 (66.2%)	<0.001
Sand floor houses	380 (92.2%)	275 (67.6%)	349 (87.3%)	<0.001
Thatched roof houses	224 (54.4%)	221 (54.3%)	98 (24.4%)	<0.001
Socio-Economic Status				
Food consumption score (Mean ± SD)	46.7 ± 27.9	38.5 ± 31.2	51.8 ± 30.7	<0.001
Poor dietary diversity	148 (35.7%)	222 (54.4%)	119 (29.8%)	<0.001

**Health Care Provision for the Sick Child**				
Health center	48 (55.2%)	41 (48.2%)	33 (48.5%)	0.241 ^NS^
Traditional medicine	34 (39.1%)	32 (37.6%)	31 (45.6%)	
None	5 (5.7%)	12 (14.1%)	4 (5.9%)	

F: female; M: male; MUAC: middle-upper arm circumference; ^NS^: non-significant result; SD: standard deviation.

**Table 2 nutrients-13-00367-t002:** Baseline demographic profile of the community health workers in the 3 study groups.

Demographic Item	Kita (High Supervision)	Kayes (Light Supervision)	Bafoulabé (No Supportive Supervision)
**Number**	90	45	34
**Sex (M/F)**	0.34	0.73	1.61
**Years of Schooling**			
9 years	6 (6.6%)	8 (17.7%)	18 (52.9%)
10–11 years	79 (87.7%)	31 (68.8%)	12 (35.2%)
12–13 years	5 (5.6%)	6 (13.4%)	4 (11.9%)
**Average Number of Years Working As Chws**			
<1 year	20 (22.2%)	8 (17.8%)	3 (8.8%)
2–4 years	14 (15.5%)	4 (8.9%)	6 (17.6%)
>4 years	56 (62.2%)	33 (73.3%)	25 (73.5%)
**Population Covered in The Catchment Area by Chws**			
Mean ± SD	1374.0 ± 665.4	1170.8 ± 499.1	1200.7 ± 747.3
<700 hab	12 (13.3%)	8 (17.7%)	14 (41. 1%)
700–1500 hab	37 (41.1%)	25 (55.6%)	15 (44.1%)
>1500 hab	41 (45%)	12 (26.7%)	5 (14.7%)
**Distance to the Health Facility**			
Mean ± SD	29.6 ±17.8	28.6 ± 17.7	31.4 ± 15.0
<5 km	1 (1.1%)	0 (0%)	0 (0%)
5–15 km	15 (16.8%)	12 (26.6%)	4 (11.7%)
16–30 km	37 (41.5%)	16 (35.5%)	15 (44.1%)
21–45 km	22 (24.7%)	15 (33.3%)	6 (17.6%)
>45 km	15 (15.7%)	2 (4.4%)	9 (26.4%)

CHWs: community health workers; F: female; hab: habitants; M: male; SD: standard deviation.

**Table 3 nutrients-13-00367-t003:** Outcomes of severe acute malnutrition treatment of children aged 6 to 59 months compared by study group and treatment providers.

Treatment Outcomes	All Districts % (95% CI)	KITA High Supervision, % (95% CI)	KAYES Light Supervision, % (95% CI)	BAFOULABÉ No Supported Supervision, % (95% CI)	Comparison Between Districts * (*p* Value)
**Cured**	81.0 (79.9–82.0)	81.4 (80.1–82.8)	86.2 (84.6–87.7)	66.9 (63.8–70.1)	^a^ < 0.001 ^b^ < 0.001; ^c^ < 0.001
CHWs	79.2 (76.9–81.5)	78.2 (74.6–81.7)	86.9 (83.3–90.4)	72.5 (67.6–77.5)	^a^ < 0.001; ^b^ 0.063; ^c^ < 0.001
Health facilities	81.4 (80.3–82.5)	82.2 (80.7–83.6)	86.1 (84.4–87.8)	63.8 (59.7–67.8)	^a^ 0.001; ^b^ < 0.001; ^c^ < 0.001
*Provider comparison (p value)*	0.090	0.031	0.724	0.009	
**Defaulted**	6.1 (5.5–6.7)	6.4 (5.6–7.2)	1.8 (1.1–2.3)	15.3 (12.9–17.7)	^a^ < 0.001; ^b^ < 0.001; ^c^ < 0.001
CHWs	6.3 (4.9–7.7)	6.6 (4.5–8.8)	5.7 (3.3–8.2)	6.4 (3.7–9.1)	^a^ 0.579; ^b^ 0.887; ^c^ 0.715
Health facilities	6.1 (5.4–6.8)	6.3 (5.4–7.2)	0.8 (0.4–1.3)	20.4 (17.0–23.7)	^a^ < 0.001; ^b^ < 0.001; ^c^ < 0.001
*Provider comparison (p value)*	0.796	0.754	<0.001	<0.001	
**Transferred**	12.5 (11.7–13.4)	11.7 (10.7–12.8)	11.9 (10.5–13.4)	16.9 (14.4–19.4)	^a^ 0.849; ^b^ < 0.001; ^c^ < 0.001
CHWs	14.3 (12.3–16.3)	14.8 (11.8–17.8)	7.4 (4.7–10.2)	21.1 (16.5–25.6)	^a^ < 0.001; ^b^ 0.019; ^c^ < 0.001
Health facilities	12.1 (11.2–13.0)	11.2 (10.0–12.3)	12.9 (11.3–14.6)	14.6 (11.6–17.5)	^a^ 0.086; ^b^ 0.024; ^c^ 0.332
*Provider comparison (p value)*	0.041	0.017	0.004	0.014	
**Death**	0.4 (0.2–0.5)	0.4 (0.2–0.6)	0.1 (0.04–0.3)	0.8 (0.2–1.4)	^a^ 0.060; ^b^ 0.186; ^c^ 0.007
CHWs	0.2 (0.06–0.4)	0.4 (0.2–0.9)	0.0	0.0	^a^ 0.249; ^b^ 0.720; ^c^-
Health facilities	0.4 (0.2–0.6)	0.4 (0.2–0.6)	0.1 (0.05–0.3)	1.3 (0.03–2.2)	^a^ 0.206; ^b^ 0.024; ^c^ 0.001
*Provider comparison (p value)*	0.338	1.000	1.000	0.108	

CHWs: community health workers; CI: confidence interval. * Comparison between districts: ^a^ Kita vs. Kayes; ^b^ Kita vs. Bafoulabé; ^c^ Kayes vs. Bafoulabé.

**Table 4 nutrients-13-00367-t004:** Average score obtained by the community health workers in the Kita and Kayes study groups for iCCM supervision of curative and preventive tasks from a maximum of 10 points.

iCCM Items	KITA (High Supervision) Mean ± SD	KAYES (Light Supervision) Mean ± SD	*p* Value
Clinical Examination of The Sick Child	9.33 ± 1.00	8.81 ± 1.37	<0.001
Newborn Monitoring	8.25 ± 2.22	8.13 ± 1.66	0.708 ^NS^
Family Planning	8.78 ± 1.96	8.28 ± 2.15	0.009
IYCF Promotion	6.20 ± 3.60	7.18 ± 3.02	0.003
Hygiene and Sanitation Promotion	8.67 ± 1.21	7.67 ± 1.81	<0.001

iCCM: integrated community case management; IYCF: infant and young child feeding; ^NS^: non-significant result; SD: standard deviation.

**Table 5 nutrients-13-00367-t005:** Outcomes of treatment provided by the community health workers of Kita and Kayes compared by the number of supervisions received for iCCM tasks and nutrition-specific tasks and the score obtained for those supervisions.

iCCM Supervision (Kita and Kayes Districts)			
Number of supervisions received (*N* = 100)	Less than 5 supervisions * (*N* = 34)	More than 5 supervisions * (*N* = 66)	Comparison
Cured % (IQR)	87.5 (66.7–100.0)	91.4 (66.7–100.0)	*p*= 0.064 ^NS^
Score obtained in the clinical examination of the sick child (*N* = 93)	Less than 9.35 points * (*N* = 48)	9.35 points or more * (*N* = 45)	Comparison
Cured % (IQR)	85.7 (66.7–100.0)	100.0 (66.7–100.0)	*p*= 0.520 ^NS^
**Nutrition Supervision (Kita District)**			
Number of supervisions received (*N* = 61)	Less than 5 supervisions * (*N* = 13)	5 supervisions or more * (*N* = 48)	Comparison
Cured % (IQR)	90.0 (56.3–100.0)	85.7 (66.7–100.0)	*p*= 0.884 ^NS^
Score obtained in the nutrition supervision (*N* = 61)	Less than 9.22 points * (*N* = 29)	9.22 points or more * (*N* = 31)	Comparison
Cured % (IQR)	83.3 (63.1–700.0)	93.3 (66.7–100.0)	*p*= 0.331^NS^

iCCM: integrated community case management; IQR: inter-quartile range; ^NS^: non-significant result; * In the iCCM supervision, the p50 of supervision was 5 and the p50 of the score obtained during the supervision was 9.35 points of 10. In the nutrition-specific supervision, the p50 of supervision was 5, and the p50 of the score obtained during the supervision was 9.22 points of 10.

## Data Availability

The data presented in this study are available on request from the corresponding author.

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
