# Peer review of "Impact of Different Levels of Supervision on the Recovery of Severely Malnourished Children Treated by Community Health Workers in Mali"

_nutrients, 2021, doi:10.3390/nu13020367_

Round 1
Reviewer 1 Report
The manuscript by Pilar Charle-Cuéllar investigated the impact of different levels of supervision on the recovery of severely malnourished children treated by community health workers in Mali. The study is informative and could help to guide the public policies in low-income countries. I have one suggestion. I think it would be more constructive if the authors could discuss in detail how the policy-makers could do to combat malnutrition based on the findings of the present study.
Author Response
It has been used "track changes", to address all of your suggestions in the manuscript. Thank you very much, really appreciate it.
Response to Reviewer 1 Comments
Point 1. The manuscript by Pilar Charle-Cuéllar investigated the impact of different levels of supervision on the recovery of severely malnourished children treated by community health workers in Mali. The study is informative and could help to guide public policies in low-income countries. I have one suggestion. I think it would be more constructive if the authors could discuss in detail how the policy-makers could do to combat malnutrition based on the findings of the present study.
Response 1. We really appreciate your suggestion. We have added at the end of the manuscript, a section of conclusion, including how the country can use this model of intervention to tackle malnutrition. Thank you very much
Changes available at the end of the manuscript
All the references have been also adapted
Response to Reviewer 2 Comments
It has been used "track changes", to address all of your suggestions in the manuscript. Thank you very much, really appreciate it. In the document below I have explained where all the changes have been made.
The study is well organized and the paper well written. Minor suggestions as follows:
Point 1. A well-structured and clear figure of study design should be inserted
Response 1. The study we have presented is a prospective non-randomized community intervention trial where all the children with acute malnutrition identify at the health facility and /or with community health workers, has been including in the analysis, we have considered not to include another figure with the design of the study. We consider that with the new information added after your suggestion in point 2, readers can have a good understanding of the study design.
Point 2.Figure 1 should be better explained in the text.
Response 2. We really appreciate your suggestion, it has been already addressed in the manuscript. We have switched the order of sections 2.2 and 2.3 and add a new one 2.5 to make them follow the same order in figure 1 and facilitate readers’ comprehension.
Changes available at:
Section 2. Material and method section.
- Subsection 2.1.paragraph 1, 2;
- subsection 2.2. paragraph 1;
- subsection 2.3. paragraph 3;
- subsection 2.4. paragraph 2,3;
- sub section 2.5. paragraph 1
- sub section 2.6. tittle
Point 3. Subparagraph 3.1. The baseline characteristics of the 3 study areas should be enlarged.
Response 3. Thank you very much, already addressed.
Changes available in Section 3. Results, paragraph 1, 2
Point 4. Data in Table 3 should be better described in the text.
Response 4. Thank you very much, already included more details.
Changes available at: Section 3, sub-section 3.2, paragraph 1, 2
Point 5. Major details should be inserted in table 4
Response 5. Thank you very much for this comment. We have already added more details related to the nutrition items and the total analyses are in appendix B. To our understanding, it could be enough.
Changes available at: Section 3, sub-section 3.3.1paragraph 2
Point 6. The section of the Conclusion should be separated by a Discussion including limits, advantages, and practical applications.
Response 6. Thank you very much. Following your suggestions, we have left limits, advantages, and practical applications in the discussion, and add a conclusion at the end of the manuscript.
Changes available at the end of the document
All the references have been also adapted

Reviewer 2 Report
The study is well organized and the paper well written. Minor suggestions as follows:
-A well structured and clear figure of study design should be inserted
-The figure 1 should be better explained in the text
-the subparagraph 3.1. Baseline characteristics of the 3 study areas should be enlarged.
-Data in Table 3 should be better described in the text.
-Major details should be inserted in table 4
The section of Conclusion should be separated by Discussion including limits, advantages and practical applications.
Author Response
It has been used "track changes", to address all of your suggestions in the manuscript. Thank you very much, really appreciate it.
Response to Reviewer 1 Comments
Point 1. The manuscript by Pilar Charle-Cuéllar investigated the impact of different levels of supervision on the recovery of severely malnourished children treated by community health workers in Mali. The study is informative and could help to guide public policies in low-income countries. I have one suggestion. I think it would be more constructive if the authors could discuss in detail how the policy-makers could do to combat malnutrition based on the findings of the present study.
Response 1. We really appreciate your suggestion. We have added at the end of the manuscript, a section of conclusion, including how the country can use this model of intervention to tackle malnutrition. Thank you very much
Changes available at the end of the manuscript
All the references have been also adapted
Response to Reviewer 2 Comments
It has been used "track changes", to address all of your suggestions in the manuscript. Thank you very much, really appreciate it. In the document below I have explained where all the changes have been made.
The study is well organized and the paper well written. Minor suggestions as follows:
Point 1. A well-structured and clear figure of study design should be inserted
Response 1. The study we have presented is a prospective non-randomized community intervention trial where all the children with acute malnutrition identify at the health facility and /or with community health workers, has been including in the analysis, we have considered not to include another figure with the design of the study. We consider that with the new information added after your suggestion in point 2, readers can have a good understanding of the study design.
Point 2.Figure 1 should be better explained in the text.
Response 2. We really appreciate your suggestion, it has been already addressed in the manuscript. We have switched the order of sections 2.2 and 2.3 and add a new one 2.5 to make them follow the same order in figure 1 and facilitate readers’ comprehension.
Changes available at:
Section 2. Material and method section.
- Subsection 2.1.paragraph 1, 2;
- subsection 2.2. paragraph 1;
- subsection 2.3. paragraph 3;
- subsection 2.4. paragraph 2,3;
- sub section 2.5. paragraph 1
- sub section 2.6. tittle
Point 3. The subparagraph 3.1. Baseline characteristics of the 3 study areas should be enlarged.
Response 3. Thank you very much, already addressed.
Changes available in Section 3. Results, paragraph 1, 2
Point 4. Data in Table 3 should be better described in the text.
Response 4. Thank you very much, already included more details.
Changes available at: Section 3, sub-section 3.2, paragraph 1, 2
Point 5. Major details should be inserted in table 4
Response 5. Thank you very much for this comment. We have already added more details related to the nutrition items and the total analyses are in appendix B. To our understanding, it could be enough.
Changes available at: Section 3, sub-section 3.3.1paragraph 2
Point 6. The section of the Conclusion should be separated by a Discussion including limits, advantages, and practical applications.
Response 6. Thank you very much. Following your suggestions, we have left limits, advantages, and practical applications in the discussion, and add a conclusion at the end of the manuscript.
Changes available at the end of the document
All the references have been also adapted
